# PSAT1 Promotes Metastasis via p-AKT/SP1/ITGA2 Axis in Estrogen Receptor-Negative Breast Cancer Cell

**DOI:** 10.3390/biom14080990

**Published:** 2024-08-12

**Authors:** Xingda Zhang, Siyu Wang, Wei Li, Jianyu Wang, Yajie Gong, Quanrun Chen, Shihan Cao, Da Pang, Song Gao

**Affiliations:** 1Department of Breast Surgery, Harbin Medical University Cancer Hospital, 150 Haping Road, Harbin 150081, China; xingdazhang@hrbmu.edu.cn (X.Z.); wangsiyu1993@126.com (S.W.); liwei@hrbmu.edu.cn (W.L.); wjy242426@163.com (J.W.); 15561803512@163.com (Y.G.); chenqrun@163.com (Q.C.); caoshihan0826@163.com (S.C.); 2Northern Translational Medical Research and Cooperation Center, Heilongjiang Academy of Medical Sciences, Harbin 150081, China

**Keywords:** cell metastasis, breast cancer, ITGA2, p-AKT/SP1/ITGA2 axis, PSAT1

## Abstract

Background: Accumulating evidence indicates that PSAT1 not only reprogrammed metabolic function but also exhibits “moonlighting” functions in promoting tumor malignancy. However, the underlying molecular mechanisms of PSAT1 promoting ER-negative breast cancer cell migration need further investigation. Methods: Briefly, the PSAT1 and ITGA2 expression in cells and tissues was detected using qRT-PCR, immunofluorescence staining and western blot assay. The effect of PSAT1 and ITGA2 was verified both in vitro and in vivo. RNA-seq analysis explored a series of differently expressed genes. The regulation between SP1 and ITGA2 was investigated by ChIP analysis. Results: We reported PSAT1 was highly expressed in ER-breast cancer tissues and tumor cells and positively correlated with metastasis. Moreover, RNA-seq analysis explored a series of differently expressed genes, including ITGA2, in PSAT1 overexpressed cells. Mechanistically, PSAT1 facilitated breast cancer metastasis via the p-AKT/SP1/ITGA2 axis. We further elucidated that PSAT1 promoted the entry of SP1 into the nucleus through the upregulation of p-AKT and confirmed ITGA2 is a target of SP1. In addition, enhanced cell migration was remarkably reversed by ITGA2 depletion or p-AKT inhibitor treatment. Conclusion: This study clarified the mechanism of PSAT1 in promoting ER-negative breast cancer metastasis, which may provide mechanistic clues for attenuating breast cancer metastasis.

## 1. Introduction

Breast cancer is one of the most common malignant tumors in women worldwide and is the second leading cause of cancer death in women [1]. Based on the molecular type of gene expression, breast cancer can be divided into intrinsic subtypes, which can be used to guide clinical treatment and prognosis [2,3,4]. The estrogen receptor (ER) is one of the most important prognostic and predictive immunohistochemical markers in breast cancer [5]. Due to the lack of effective endocrine therapy, patients with ER-negative breast cancer usually have a worse prognosis and shorter survival time than patients with ER-positive breast cancer [6]. Importantly, treatments for metastatic patients are currently palliative, and they are still ineffective for metastatic breast cancer [7]. As a result, it is critical to investigate the mechanism of ER-negative breast cancer metastasis and identify key molecular targets.

There is accumulating evidence that the serine synthesis pathway (SSP) is activated during tumorigenesis as well as in metastatic breast cancer, which is an essential metabolic reprogramming pathway [8,9]. Phosphoserine aminotransferase 1 (PSAT1), one of the key enzymes in the SSP, catalyzes the conversion of 3-phosphohydroxy-pyruvate to 3-phosphoserine and plays an important role in tumor development [8,10,11,12]. PSAT1 was identified as an oncogene in earlier research and is associated with metastasis and poor prognosis in many malignancies, including lung cancer, liver cancer, and colorectal cancer [13,14,15].

Our previous work revealed that PSAT1 overexpression activates the GSK-3β/β-catenin signaling pathway, facilitating the entry of the transcription factor β-catenin into the nucleus and promoting the proliferation of ER-negative breast cancer cells [16]. A recent study found that PSAT1 also activates the Notch1/β-catenin signaling pathway to upregulate the metastatic ability of breast cancer and promote distant metastasis [17]. Similarly, PSAT1 interacts with IQGAP1 and, subsequently, promotes cell migration through STAT3 phosphorylation [18]. We wondered whether PSAT1 could extensively activate key proteins affecting malignant functions in tumors through phosphorylation. AKT, a well-known protein regulating GSK-3β, is frequently overactivated by phosphorylation in malignant tumors [19]. Moreover, AKT inhibitors are potentially effective drugs for the treatment of breast cancer [20]. Therefore, we propose that PSAT1 may promote metastasis in ER-negative breast cancer by regulating AKT phosphorylation. We wondered whether PSAT1 is capable of broadly activating key proteins affecting malignant tumor function through phosphorylation and, ultimately, promoting breast cancer metastasis.

In this study, we explored whether PSAT1 was overexpressed in ER-negative breast cancer and was related to tumor metastasis. PSAT1-overexpressing ER-negative breast cancer cells were analyzed using RNA-seq analysis, and a series of differentially expressed genes, including ITGA2, were identified. ITGA2 has been demonstrated to play a significant role in tumor cell proliferation and metastasis [21,22,23,24]. Furthermore, our results revealed that the expression of SP1, the upstream transcription factor of ITGA2 [25,26,27], was upregulated by PSAT1 to promote ITGA2 expression. Taken together, our findings suggest that PSAT1 boosts ER-negative breast cancer metastasis via the p-AKT/SP1/ITGA2 axis.

## 2. Materials and Methods

### 2.1. Reagents

All the reagents used in this study are listed in Appendix A.

### 2.2. Cell Lines and Culture

The human breast cancer cell lines (BT-549 and HCC1937) and the mice breast cancer cell lines(4T1) were obtained from the Procell Life Science and Technology Co., Ltd. (Wuhan, China). All cell lines were authenticated using short tandem repeat (STR) profiling within the last 3 years. All experiments were performed with mycoplasma-free cells. The 4T1 cells were cultured in basic DMEM (Gibco, NY, USA) containing 10% fetal bovine serum and 1% penicillin/streptomycin. BT-549 and HCC1937 cells were cultured in a basic RPMI-1640 medium (Gibco, NY, USA) containing 10% fetal bovine serum and 1% penicillin/streptomycin. All cells were maintained in a humidified 5% CO_2_ atmosphere at 37 °C.

### 2.3. Cell Transfection and Lentiviruses Infection

Lentiviral construct and packing plasmid were purchased from GeneChem (Shanghai, China). For lentiviral transfection, refer to the Lentiviral Transfection Handbook for Shanghai Genechem. For plasmid and siRNAs transfection, PSAT1 plasmid (Clontech, Shanghai, China), PSAT1 and ITGA2 siRNAs (RiboBio, Guangzhou, China) were transfected into cells using the jetPRIME (Polyplus, Illkirch, France) reagent according to the manufacturer’s protocol. The final concentration of the plasmid used was 5 μg/mL, and the siRNA transfection concentration was 50 nM. The sequences for lentivirus and all siRNAs are listed in Appendix A.

### 2.4. Phenotypic Experiment

For the wound healing assay, the bottom of the well plate was scraped when cell fusion reached 100%. For migration/invasion experiments, cells in a serum-free medium were inoculated into transwell chambers (Corning. Corning, NY, USA), which were additionally coated with Matrigel (Corning) in the invasion assay. The bottom chamber was filled with a 20% FBS (700 μL) medium and incubated at 37 °C for 24 h or 36 h for migration or invasion experiments. For cell adhesion experiments, cells were recounted and inoculated in 96-well plates at a density of 10,000 cells per well. Formaldehyde fixation was performed after 90 min incubation at 37 °C. All phenotyping experiments were stained using crystal violet for 15 min, photographed under a microscope, and ImageJ was used for calculations.

### 2.5. Patients and Tissue Specimens

The tissue samples included in this study were from the pathology department and breast cancer biobank [28] of Harbin Medical University Cancer Hospital. Eligible patients were those with a histological diagnosis of cancer who had received neither adjuvant chemotherapy, immunotherapy, nor radiotherapy before surgery. The patients with recurrent tumors, metastatic disease, bilateral tumors, or other previous tumors were excluded. Pathologists examined tumors for confirmation of the diagnostically and molecular subtype of each cancer tissue. This study was approved by the Ethical Committees of Harbin Medical University Cancer Hospital. We obtained written informed consent from all subjects who agreed to donate specimens to the biobank.

### 2.6. Immunohistochemistry

IHC was used to detect the PSAT1 protein expression of breast cancer tissues, which was performed using the same procedure as before [16]. Primary antibodies were incubated overnight at 4 °C using anti-PSAT1 antibody (Genetex, Irvine, CA, USA). Microscopic photographs were taken for scoring.

### 2.7. Immunofluorescence Staining

Cells were blocked using goat serum, incubated overnight at 4 °C with a primary antibody, incubated at room temperature for 2 h with a secondary antibody, and stained with DAPI. The primary antibody used was PSAT1 (Genetex, 1:100) and ITGA2 (Santa Cruz Biotechnology, Santa Cruz, CA, USA, 1:100). Images were acquired using a fluorescence microscope and analyzed using ImageJ software (version number: ImageJ 1.53).

### 2.8. Western Blotting and Antibodies

For total protein extraction, cells were lysed using RIPA buffer containing protease inhibitor and phosphatase inhibitor. For cytosolic nuclear separation, proteins used were from the Nuclear and Cytoplasmic Protein Extraction Kit (Wanleibio, Shenyang, China), and the operation was carried out in strict accordance with the product instructions. A western blotting assay was performed as previously described [16]. The following antibodies were used: anti-PSAT1 (Genetex, 1:1000), ITGA2 (Abcam, UK, 1:2000), anti-p-AKT (Cell Signaling Technology, Danvers, MA, USA, 1:1000), anti-SP1 (Proteintech, Wuhan, China, 1:1000) anti-β-actin (Origene, Shanghai, China, 1:1000), and anti-β-Tubulin (Proteintech, 1:1000).

### 2.9. Animal Experiments

All experimental procedures involving animals were performed following the animal protocols approved by the Medical Laboratory Animal Care Committee of Harbin Medical University. Five-week-old wild-type BALB/C mice were obtained from Beijing Weitehe Laboratory Animal Technology Company and housed in a temperature-controlled environment. BALB/c mice and constructed 4T1 cells were used for the establishment of the metastasis model. Then, about 1 × 10^5^ cells (4T1-PSAT1-NC, 4T1-PSAT1-KD1, 4T1-PSAT1-KD2) 0.7 × 10^5^ cells (4T1-PSAT1-CON, 4T1-PSAT1-OE) in a 100 μL DMEM medium were injected into the tail vein to establish a lung metastasis model. After 27 days of injection, mice were euthanized by cervical dislocation, and lung tissue was taken and then fixed with 4% paraformaldehyde. After paraffin embedding, tissue sectioning was performed.

### 2.10. RNA Isolation and qRT-PCR

Total RNA samples from cell samples were extracted using Trizol reagent (Invitrogen, Carlsbad, CA, USA) according to the manufacturer’s protocol. Total RNA was then reverse transcribed using Transcriptor Fast Strand cDNA Synthesis Kit (TIANGEN Biotechnology, Beijing, China) to obtain complementary DNA (cDNA). mRNA expression was examined with real-time PCR using FastStart Universal SYBR Green Master (ROX) with gene-specific primers and an ABI StepOne real-time PCR system. For the quantification of gene expression, we used the 2^−ΔΔCT^ method. β-actin was used as a control. All primer sequences are listed in Appendix A.

### 2.11. Chromatin Immunoprecipitation

Chromatin immunoprecipitation (ChIP) assays were performed using the ChIP Assay Kit (Beyotime) according to the manufacturer’s protocol. The chromatin DNA was sonicated and sheared to 100–200 bp fragments. Stained chromatin was cultured overnight with an SP1(Genetex) or IgG serving as the negative control. The DNA was subjected to PCR to amplify the ITGA2 binding sites. The amplified fragments were then analyzed on an agarose gel. Chromatin (1%), prior to immunoprecipitation, was used as the input control. The primer sequence was as follows: 5′-GTTTGCATCCCTGCGTGT-3′ and 5′-CCGAGCTTCCTCACCAACT-3′.

### 2.12. RNA-Sequencing Analysis

RNA preparation, library construction, and sequencing were carried out on the BGISEQ-500 platform at the Beijing Genomics Institute (BGI, Shenzhen, China). Statistical analysis was performed, and differentially expressed genes (DEGs) were selected that met the criteria of a fold change  ≥ 1.5 and *p* ≤  0.05. RNA sequencing data are available from the corresponding author.

### 2.13. Public Data Access

The Cancer Genome Atlas (TCGA)-BRCA gene expression profiles in the fragments per kilobase million (FPKM) format were downloaded from the TCGA databases using the R package “TCGAbiolinks”. PSAT1 subtype cells for breast tumors and cell lines were obtained from UCSC Xena [29]. All transcripts were normalized using log^2^ transformation. PSAT1 expression was dichotomized using a study-specify median expression as the cutoff to define “high expression” as an expression level at or above the median versus “low expression” as an expression level below the median.

### 2.14. Statistics

Statistical analysis was performed using GraphPad Prism 9 software and Excel. Statistical significance was assessed based on the number of groups with one or more independent variables. The number of mice in each group has been indicated for specific in vivo experiments. All the experiments were performed in triplicate, and all of the statistical tests were two sided. Data were shown as means ± SD. A *p* value of <0.05 was considered statistically significant for all analyses.

## 3. Results

### 3.1. PSAT1 Is Overexpressed in ER-Negative Breast Cancer with Lymph Node Metastasis

To investigate the role of PSAT1 in breast cancer, we first analyzed its mRNA expression in breast cancer using the TCGA database. The results showed that PSAT1 expression was downregulated in breast cancer tissues compared to normal tissues (Figure 1A). However, PSAT1 expression was significantly upregulated in ER-negative breast tissue compared to that in ER-positive breast tissue (Figure 1B). Meanwhile, survival analysis revealed that PSAT1 expression in ER-negative breast cancer was significantly associated with patient survival (Appendix A). Next, we evaluated PSAT1 expression in 360 breast cancer tissues to clarify the clinical significance of PSAT1 expression. The PSAT1 immunohistochemical (IHC) staining intensity was positively correlated with the ER status (Figure 1C,D). Furthermore, when we divided these ER-negative clinical samples into nonmetastatic (*n* = 83) and metastatic (*n* = 65) groups based on lymph node status, more metastatic samples (33/65) had intense PSAT1 cytoplasmic staining than did nonmetastatic samples (28/83) (Figure 1E). However, in ER-positive breast cancer tissues, there was no significant difference in PSAT1-positive expression between the two groups (Figure 1F). Accordingly, we hypothesized that PSAT1 levels are correlated with the occurrence of node metastasis in ER-negative breast cancer patients. As expected, most ER-negative breast cancer cell lines exhibited higher levels of PSAT1 expression than did the ER-positive cell lines (Appendix A). Overall, we found that PSAT1 was overexpressed in ER-negative breast cancer and correlated with metastasis.

### 3.2. PSAT1 Facilitates the Metastasis of ER-Negative Breast Cancer Cells

To explore the potential regulatory mechanism of PSAT1 in breast cancer, we knocked down or overexpressed PSAT1 by lentiviral transduction in the ER-negative breast cancer cell lines BT-549 and HCC1937. The expression levels of the PSAT1 protein and mRNA were detected via WB and qRT-PCR (Figure 2A,B). Cell adhesion and Transwell experiments showed that downregulation of PSAT1 significantly reduced the adhesion and migration capacity of BT-549 and HCC1937 cells compared to control treatments (Figure 2C). The same results were supported by wound healing experiments (Figure 2D,E). On the contrary, transwell experiments revealed that overexpression of PSAT1 significantly promoted cell adhesion, invasion, and migration compared to the control group (Figure 2F). Additionally, scratch assays yielded the same results (Figure 2G,H). Next, PSAT1 was knocked down and overexpressed in BT-549 cells using siRNA and plasmids (Appendix A). PSAT1 deletion significantly inhibited the cellular metastatic capacity of the cells (Appendix A). In contrast, the overexpression of PSAT1 enhanced cell adhesion, migration, and invasion. Collectively, these results indicate that aberrant upregulation of PSAT1 is required for ER-related breast cancer cell metastasis.

### 3.3. PSAT1 Promotes the Metastasis of ER-Negative Breast Cancer Cells In Vivo

To validate the role of PSAT1 in controlling cancer metastasis in vivo, we generated 4T1 cells with stable knockdown or overexpression of PSAT1 (murine-derived breast cancer cell line). The expression levels of PSAT1 in the corresponding cells were verified (Figure 3A). Similarly, PSAT1 deletion in 4T1 cells also reduced cell adhesion, migration, and invasion (Figure 3B). In contrast, PSAT1-OE cells exhibited enhanced adhesion, migration, and invasion capacity in vitro (Figure 3B). The findings revealed that mice transplanted with 4T1-KD1 or 4T1-KD2 cells had considerably fewer lung metastases than mice transplanted with 4T1-NC cells (Figure 3C). Conversely, mice transplanted with 4T1-OE cells exhibited significantly more lung metastases than mice transplanted with 4T1-CON cells (Figure 3D). Taken together, these findings suggested that PSAT1 promotes ER-negative breast cancer metastasis in vivo.

### 3.4. PSAT1 Enhances ER-Negative Breast Cancer Metastasis through the Upregulation of ITGA2

To further explore the molecular mechanisms underlying the enhanced metastasis of PSAT1 in ER-negative breast cancer, we performed RNA sequencing (RNA-seq) on PSAT1-overexpressing BT-549 cells (Figure 4A and Appendix A). A Venn diagram was constructed, which showed that 328 candidate genes were potential downstream targets of PSAT1 (Appendix A). A heatmap representing DEGs was constructed, where the threshold was set at |fold change| >1.5 and a *p* value < 0.05. (Appendix A). Functional annotation was accomplished using gene ontology (GO) enrichment (Appendix A), and KEGG pathway analysis (Figure 2B and Appendix A) was used to investigate the biological role of DEGs induced by PSAT1. DEGs were intimately associated with cell migration and cell adhesion and were enriched in the PI3K-Akt signaling pathway. We screened seven metastasis-related DEGs as candidate genes [23,30,31,32,33,34,35] and verified the differential expression levels of these genes by RT-PCR (Figure 2C). ITGA2 was significantly upregulated in PSAT1-overexpressing cells and was therefore selected for further study.

To elucidate the mechanism of PSAT1-mediated ITGA2 regulation, we first detected ITGA2 expression in PSAT1-knockdown or -overexpressing cells. WB and qRT-PCR experiments demonstrated that ITGA2 mRNA and protein expression was significantly lower in PSAT1-knockdown BT549 and HCC1937 cells than in control cells. Conversely, ITGA2 expression was correspondingly increased in PSAT1-overexpressing BT549 and HCC1937 cells but not in control cells (Figure 4D,E and Appendix A). Consistent findings were also obtained for the murine breast cancer cell Line 4T1 (Appendix A). Furthermore, immunofluorescence assays revealed that silencing PSAT1 reduced the intensity of ITGA2 fluorescence, whereas overexpressing PSAT1 enhanced its intensity (Figure 4F). Subsequently, we silenced ITGA2 in PSAT1-overexpressing BT-549 cells (Figure 4G). As expected, adhesion, transwell, and wound healing assays showed that ITGA2 depletion markedly reversed the increase in cell metastasis caused by PSAT1 overexpression (Figure 4H–J). These findings revealed that PSAT1 promotes cell metastasis by upregulating ITGA2 expression.

### 3.5. PSAT1 Regulated ITGA2 Expression through SP1

The transcription factor SP1 can reportedly positively regulate ITGA2 [25]. In this study, we initially investigated the potential for PSAT1 to enhance the modulation of SP1. As depicted in Figure 5A, our results demonstrated that overexpression of PSAT1 resulted in an upregulation of SP1 expression within the nucleus. However, more SP1 protein was enriched in the cytoplasm when PSAT1 was silenced (Figure 5A). To further verify the regulatory relationship between SP1 and ITGA2, we first used the JASPAR (http://jaspar.genereg.net/, accessed on 27 June 2022) website to predict the binding site of SP1 to ITGA2 (Appendix A). ChIP assays suggested that PSAT1 overexpression increased the binding of SP1 to the ITGA2 promoter, which confirmed that SP1 is a transcription factor for ITGA2 (Figure 5B,C). These results suggest that PSAT1 promotes the translocation of SP1 from the cytoplasm to the nucleus and facilitates its function. Next, silencing of SP1 with mithramycin (MIT), an SP1 inhibitor, reduced the mRNA and protein expression of ITGA2 compared to that in the control groups (Figure 5D,E). In addition, the SP1 inhibitor reduced the extent of cell metastasis caused by PSAT1 overexpression (Figure 5F–H). In conclusion, the above findings suggest that SP1 can bind directly to the ITGA2 promoter and activate its transcription, which can then be regulated by PSAT1 in ER-negative breast cancer cells and eventually lead to metastasis.

### 3.6. PSAT1-Regulated Tumor Metastasis via the p-AKT/SP1/ITGA2 Axis

Since KEGG pathway enrichment analysis revealed that most of the DEGs induced by PSAT1 overexpression were enriched in the PI3K/AKT signaling pathway, we hypothesized that PSAT1 could regulate SP1 through the PI3K/AKT signaling pathway. As shown in Figure 6A, overexpressed PSAT1 stimulated p-AKT protein expression, while PSAT1 silencing downregulated p-AKT protein expression in BT-549 and HCC1937 cells. To further confirm whether p-AKT acts as a shutdown factor for the PSAT1-mediated regulation of SP1/ITGA2, we used LY294002 (a PI3K-AKT pathway inhibitor) to suppress p-AKT expression (Figure 6B). Immunoblotting revealed that PSAT1 overexpression activated the expression of ITGA2, which was reversed in the LY294002-treated group compared with the control group, suggesting that the upregulation of ITGA2 expression caused by PSAT1 is dependent on the expression level of p-AKT (Figure 6C). As expected, the administration of the inhibitor LY294002 also effectively decreased the level of the SP1 protein in the nucleus (Figure 6C). Moreover, the increase in ITGA2 mRNA expression in PSAT1-OE cells was attenuated by the inhibitor LY294002 (Figure 6D). In addition, downregulation of p-AKT expression by the inhibitor LY294002 abolished the increase in cell metastasis in PSAT1-OE cells compared to that in control cells (Figure 6E,F). Taken together, these results provide solid evidence that PSAT1 promotes cell metastasis through the p-AKT/SP1/ITGA2 axis in ER-negative breast cancer cells (Figure 6F).

## 4. Discussion

PSAT1 is one of the key enzymes of the serine synthesis pathway and is thought to play an important role in the regulation of gene expression, DNA repair, cell proliferation, and metastasis in tumors [15,16,36,37], in addition to its role in the metabolic process of tumor cells. In previous studies, PSAT1 was shown to be associated with the development and metastasis of a variety of cancers, including lung, liver, ovarian, colorectal, and esophageal cancer [14,36,38,39,40]. Here, we demonstrated that PSAT1 is significantly upregulated in ER-negative breast cancer cells and induces distant metastasis in breast cancer cells. Silencing of PSAT1 in ER-negative breast cells significantly reduces the migratory invasive capacity of the cells. In contrast, PSAT1 overexpression upregulated the adhesion, migratory, and invasive abilities of breast cancer cells and increased the distant metastasis of tumors in a mouse model.

According to previous reports, the high expression of PSAT1 in breast cancer can promote proliferation and metastasis [40,41]. In breast cancer cells with bone metastases, the SSP pathway is promoted by upregulating PSAT1 to increase glutamine synthesis and decrease glucose dependence [42]. In addition, SSP has been associated with drug resistance in ER-positive breast cancer, where increased serine synthesis reduces the sensitivity of tumor cells to tamoxifen, leading to drug resistance in tumor cells [43]. Our previous study revealed that increased PSAT1 expression in ER-negative breast cancer induces phosphorylation of GSK3β [16]. In this work, we further investigated the downstream regulatory mechanism of PSAT1 and proposed a new downstream regulatory pathway of PSAT1 that is not dependent on the SSP. PSAT1 could increase distant breast cancer metastasis through upregulation of the p-AKT/SP1/ITGA2 axis. PSAT1 was highly expressed in ER-negative breast cancer tissues, and a retrospective analysis revealed that high PSAT1 expression was significantly associated with lymph node metastasis. To further determine the relationship between PSAT1 and metastasis, we verified that the overexpression of PSAT1 promoted the adhesion, migration, and invasion of ER-negative breast cancer cells. The overexpression of PSAT1 promoted the formation of lung metastasis.

In the present study, we identified differentially expressed genes, including ITGA2, by transcriptome sequencing of PSAT1-overexpressing breast cancer cell lines. Early studies have shown that ITGA2 is overexpressed in a variety of tumors and is closely associated with tumor cell proliferation, migration, and invasion [44,45]. In this study, we confirmed that PSAT1 significantly promoted ITGA2 expression. Furthermore, silencing ITGA2 abrogates the promotion of tumor cell migration, invasion, and adhesion caused by PSAT1 overexpression. Therefore, we suggest that PSAT1 affects breast cancer metastasis by regulating the expression of ITGA2. It has been suggested that SP1 can bind to ITGA2 [25]. PSAT1 overexpression promotes SP1 protein expression in the nucleus and a decrease in the cytoplasm. In contrast, PSAT1 knockdown significantly inhibited SP1 entry into the nucleus. We confirmed that SP1 acts as a transcription factor that directly binds to the ITGA2 promoter region, ultimately promoting its transcriptional activation.

The serine/threonine protein kinase AKT (also known as protein kinase B, or PKB) is one of the most commonly overexpressed kinases in human cancers and regulates a variety of cellular processes, including cell proliferation, survival, metabolism, growth, invasion, and angiogenesis [46]. Increasing evidence suggests that activation of AKT and its related pathways is highly correlated with tumorigenesis, with P-AKT being the main mode of AKT activation [47,48]. KEGG pathway enrichment analysis indicated that PSAT1 might be involved in the regulation of the AKT pathway. We suggest that the overexpression of PSAT1 promotes the phosphorylation of AKT and increases the entry of SP1 into the nucleus, which transcriptionally activates ITGA2 and, ultimately, enhances the migration and invasion of ER-negative breast cancer cells. However, the inhibition of PSAT1 led to the opposite result. Thus, the PSAT1/p-AKT/SP1/ITGA2 axis may be the molecular mechanism that promotes metastasis in ER-negative breast cancer cells.

In summary, this study clarified the regulatory relationship between PSAT1 and ITGA2 for the first time and preliminarily explored its regulatory mechanism. For the upstream regulation of PSAT1, we previously reported that PSAT1 can be transcriptionally activated by the transcription factor ATF4 in breast cancer [16]. In addition, PSAT1 can also be regulated by the degree of DNA methylation of its promoter, which is negatively correlated with its expression at the mRNA level [49]. PSAT1 is affected by DNA methylation in ER-positive breast cancer, resulting in its low expression [50]. Meanwhile, our data also proved that the relationship between PSAT1 and metastasis in ER-positive breast cancer was not significant, but whether there are other regulatory pathways deserves further investigation. In conclusion, our study provides a new theoretical basis for the regulatory role of PSAT1 in tumors, which is of great significance for the individualized treatment of breast cancer.

## 5. Conclusions

In this study, we found that PSAT1 is significantly highly expressed in ER-negative breast cancer and was able to promote breast cancer cell metastasis by regulating the ITGA2 protein. Furthermore, PSAT1 enhances SP1 expression in the nucleus by the upregulation of p-AKT. Finally, knockdown PSAT1 or the administration of LY294002 or MIT decreased the expression of ITGA2 and abolished the enhanced cell metastasis, which provides a promising strategy for ER-negative breast cancer metastasis.

## Figures and Tables

**Figure 1 biomolecules-14-00990-f001:**
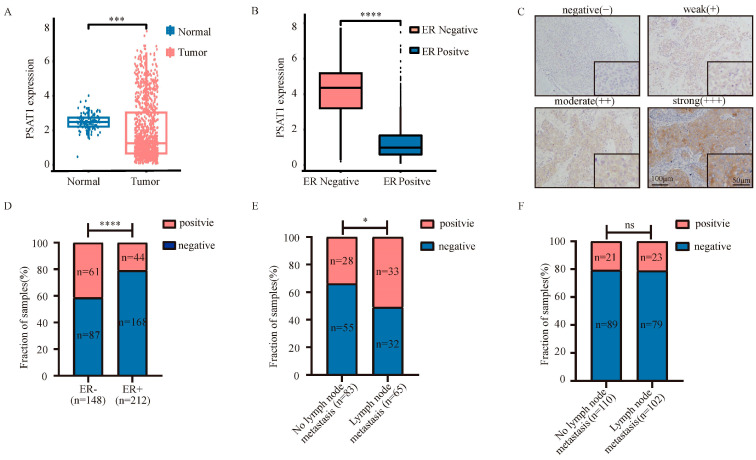
PSAT1 overexpressed in ER negative breast cancer with lymph node metastasis. (**A**) Expression profile of PSAT1 in primary breast cancer tissues (*n* = 1113) compared with normal breast tissues (*n* = 113) (TCGA). (**B**) Expression profile of PSAT1 in ER− breast cancer tissues (*n* = 238) compared with ER+ tissues (*n* = 808) (TCGA). (**C**) Representative images of PSAT1 immunohistochemical staining in breast cancer samples; scale bar, 100 µm. (**D**) Quantification of positive or negative PSAT1 expression in ER− or ER+ BC samples by Chi-square test. Quantification of positive or negative PSAT1 expression in ER− (**E**) and ER+ (**F**) BC samples with corresponding LN status by Chi-square test. TCGA = The Cancer Genome Atlas; ER− = ER negative; ER+ = ER positive; lymph node = LN; BC = breast cancer. * *p* < 0.05, *** *p* < 0.001, **** *p* < 0.0001, ns: no significance.

**Figure 2 biomolecules-14-00990-f002:**
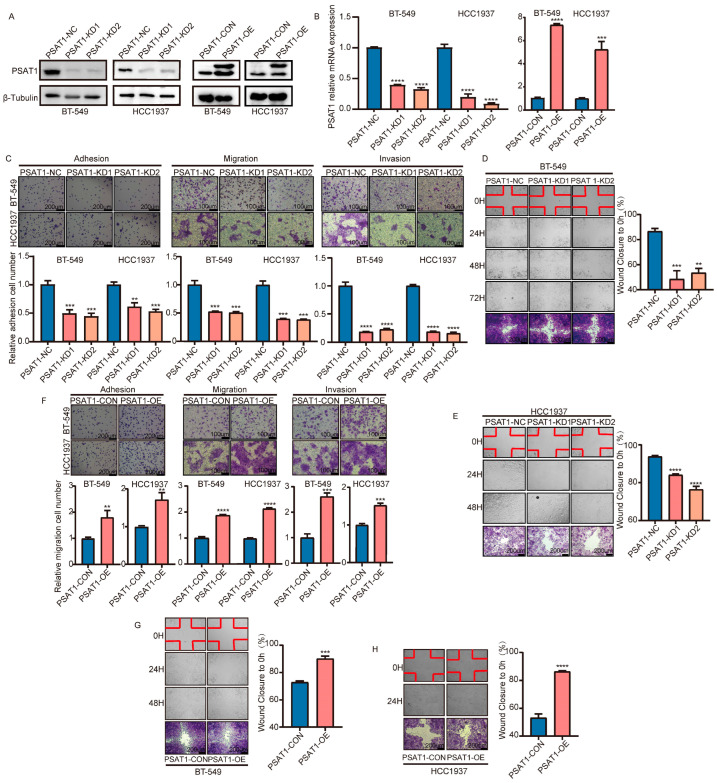
PSAT1 knockdown prevented metastasis of ER-negative breast cancer cells. Western blot (**A**) and qRT-PCR assay (**B**) showed PSAT1 knockdown and overexpression in BT-549 and HCC1937 cells infected with PSAT1 lentivirus (PSAT1-KD1, PSAT1-KD2 and PSAT1-OE) or control (PSAT1-NC and PSAT1-CON). The values of the PSAT1-NC and PSAT1-CON groups were normalized to 1. (**C**,**F**) Image and quantification of adhesion assay, transwell migration, and invasion assays in BT-549 and HCC-1937 cells. The cell numbers of PSAT1-NC and PSAT1-CON group were normalized to 1; scale bar for adhesion assay 200 µm and for transwell migration and invasion assay 100 µm. The wound healing assay revealed PSAT1 knockdown inhibited cell metastasis in BT-549 (**D**) and HCC1937 (**E**) cellsThe wound healing assay revealed PSAT1 overexpression promoted cell metastasis in BT-549 (**G**) and HCC1937 (**H**) cells; scale bar, 200 µm. Statistical analysis was performed using unpaired two-tailed Student *t*-test. For (**C**–**H**), the results are expressed as the mean ± SD; *n* = 3. ** *p* < 0.01, *** *p* < 0.001, **** *p* < 0.0001.

**Figure 3 biomolecules-14-00990-f003:**
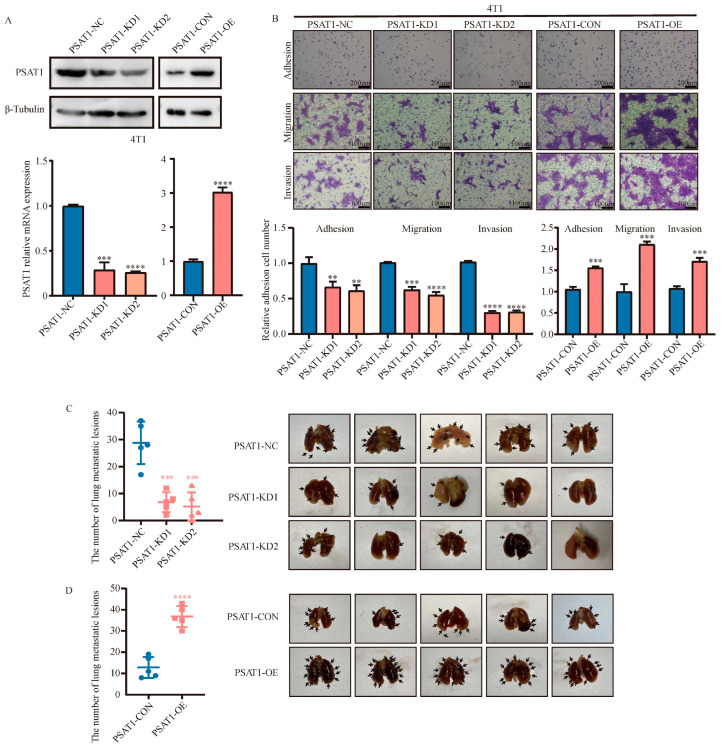
PSAT1 promoted metastasis of ER-negative breast cancer cells in mouse model. Western blot (**A**) and qRT-PCR (**B**) were used to analyze PSAT1 expression in PSAT1-knockdown (PSAT1-KD1, PSAT1-KD2) or control (PSAT1-NC) 4T1 cells and PSAT1-overexpression (PSAT1-OE) or vector (PSAT1-CON) 4T1 cells. The adhesion assays, transwell migration assays, or invasion assays (**B**) showed the cellular transfer ability of the indicated cells. The number of PSAT1-NC or PSAT1-CON cells were normalized to 1. Scale bar for adhesion assays were 200 µm and for transwell migration assays or invasion assays were 100 µm. Images and quantification of BALB/C mice tail vein injection lung metastasis mode with PSAT1 knockdown (**C**) and PSAT1 overexpression (**D**) 4T1 cells. The quantification was analyzed using Student’s *t*-test for comparisons. For (**B**), the results are expressed as the mean ± SD; *n* = 3. Lung tissues were resected from mice at 27 days. Lung metastases were counted. (*n* = 5). ** *p* < 0.01, *** *p* < 0.001, **** *p* < 0.0001.

**Figure 4 biomolecules-14-00990-f004:**
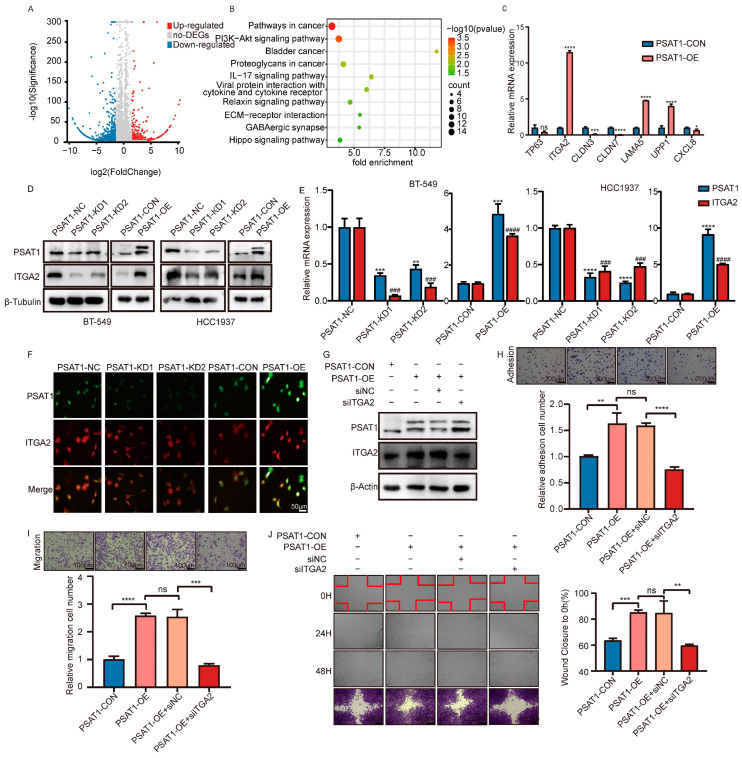
PSAT1 promoted metastasis of ER-negative breast cancer cells through upregulation of ITGA2 (**A**). The volcano maps of all samples. Red dots mean the signal value of up-regulated genes (*n* = 1276), and blue ones indicate the signal value of down-regulated genes (*n* = 535) (**B**). Bubble map of KEGG enrichment analysis for DEGs (**C**). qRT-PCR is used to validation in the indicated BT-549 cells. The PSAT1 ITGA2 expression was analyzed with western blot (**D**) and qRT-PCR (**E**) in the indicated cells. The values of the PSAT1-NC and PSAT1-CON groups were normalized to 1 (**F**). Immunofluorescence staining showed PSAT1 knockdown reduced ITGA2 expression, but PSAT1 overexpression promoted ITGA2 expression. Scale bar, 50 µm (**G**). Immunoblot assay of PSAT1 and ITGA2 protein levels in PSAT1-CON, PSAT1-OE, PSAT1-OE+siNC, and PSAT1-OE+siITGA2. The adhesion assay (**H**) and transwell migration assay (**I**) of BT-549 cells with PSAT1-CON, PSAT1-OE, PSAT1-OE+siNC, and PSAT1-OE+siITGA2. The cell numbers of PSAT1-CON groups were normalized to 1. Scale bar, 100 µm (**J**). Wound healing assay showed that silencing ITGA2 abrogates cell metastasis due to PSAT1 overexpression. Scale bar, 200, µm. Statistical analysis was performed with unpaired two-tailed Student *t*-test. For (**H**–**J**), the results are expressed as the mean ± SD; *n* = 3. #ITGA2 level compared to PSAT1-NC or PSAT1-CON. * *p* < 0.05, ** *p* < 0.01, *** *p* < 0.001, **** *p* < 0.0001, ### *p* < 0.001, #### *p* < 0.0001, ns: no significance. DEGs: differentially expressed genes.

**Figure 5 biomolecules-14-00990-f005:**
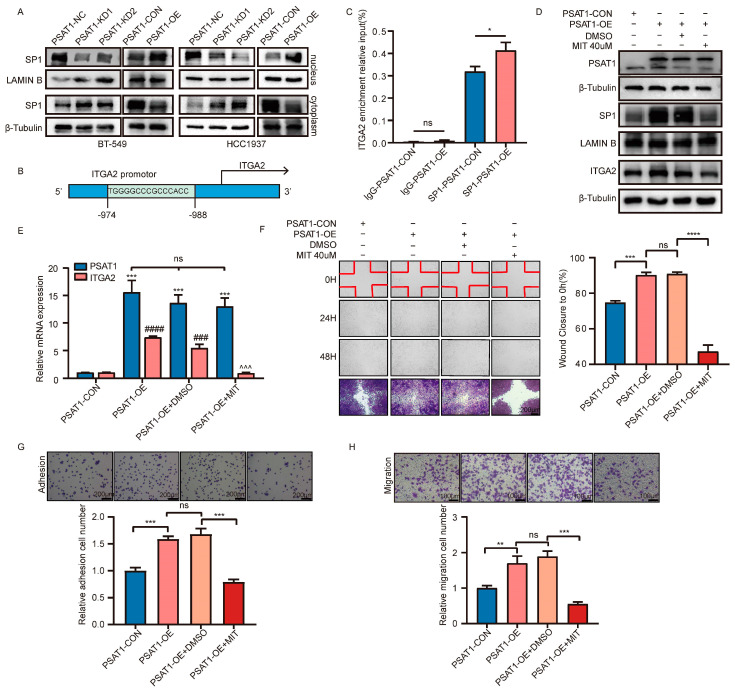
PSAT1 upregulated ITGA2 expression through transcription factors SP1 (**A**). Western blot was performed to analyze the expression of SP1 in nucleus and cytoplasm of indicated cells (**B**). Schematic representation of the predicated SP1 binding site within the ITGA2 promotor (**C**). Binding of SP1 to the ITGA2 promoter region in vitro was assessed using ChIP with anti-SP1 or anti-IgG antibodies in BT-549 cells. Input DNA purified by ChIP assay were measured using qRT-PCR. The results of IgG were normalized to 1 (**D**). Western blot was used to show PSAT1 and ITGA2 protein levels in PSAT1-CON, PSAT1-OE, PSAT1-OE+DMSO, and PSAT1-OE+MIT (**E**). qRT-PCR quantification of the indicated mRNAs in BT-549 cells (**F**). Wound healing assay showed that silencing SP1 abrogates cell metastasis caused by PSAT1 overexpression. Scale bar, 200 µm. The adhesion assay (**G**) and transwell migration assay (h) of BT-549 cells with PSAT1-CON, PSAT1-OE, PSAT1-OE+DMSO, and PSAT1-OE+MIT. Scale bar, 100 µm. Statistical analysis was performed with unpaired two-tailed Student *t*-test. For (**D**–**H**), treated with MIT (40 μM) for 24 h. For (**F**–**H**), the results are expressed as the mean ± SD; *n* = 3. For *p* values in e *PSAT1 level compared to PSAT1-CON; #ITGA2 level compared to PSAT1-CON; ^ITGA2 level compared to PSAT1-OE+DMSO. * *p* < 0.05, ** *p* < 0.01, *** *p* < 0.001, **** *p* < 0.0001, ### *p* < 0.001, #### *p* < 0.0001, ^^^ *p* < 0.001, ns: no significance.

**Figure 6 biomolecules-14-00990-f006:**
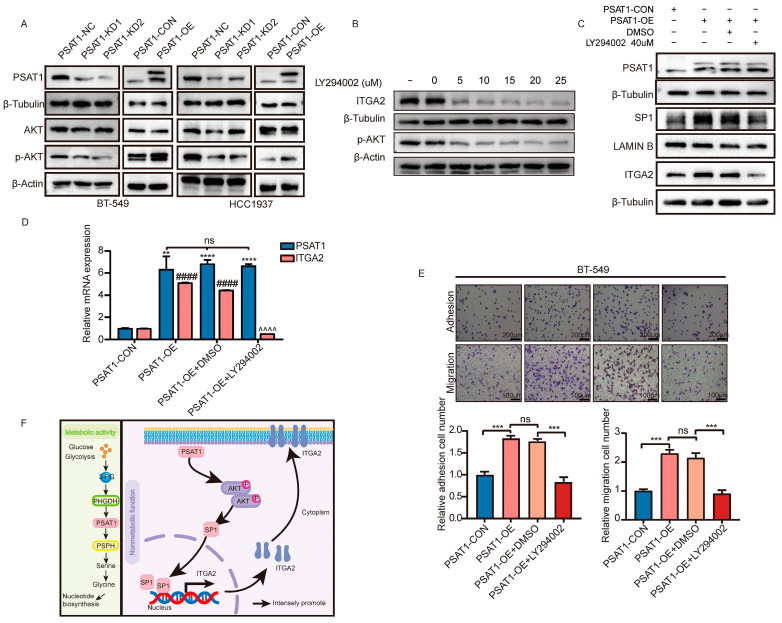
PSAT1 promoted metastasis of ER-negative breast cancer cells by p-AKT/SP1/ITGA2 axis (**A**). Western blot was performed to analyze the expression of PSAT1 and AKT/p-AKT in the indicated cells (**B**). PSAT1 overexpression cell treated with PI3K-AKT pathway inhibitor. Then the expression of P-AKT and ITGA2 were tested (**C**). Western blot was used to show PSAT1, SP1, and ITGA2 protein levels in PSAT1-CON, PSAT1-OE, PSAT1-OE+DMSO, and PSAT1-OE+LY294002 (**D**). qRT-PCR quantification of the indicated mRNAs in BT-549 cells (**E**). The adhesion assay and transwell migration assay of BT-549 cells showed that silencing p-AKT pathway abrogates cell metastasis caused by PSAT1 overexpression. Statistical analysis was performed using unpaired two-tailed Student *t*-test (**F**). Proposed model for PSAT1 promotes estrogen receptor negative breast cancer cell metastasis via p-AKT/SP1/ITGA2 pathway. For (**C**–**E**), treated with LY294002 (10 μM) for 1 h. For (**E**), the results are expressed as the mean ± SD; *n* = 3. For *p* values in e *PSAT1 level compared to PSAT1-CON; #ITGA2 level compared to PSAT1-CON; ^ITGA2 level compared to PSAT1-OE+DMSO. ** *p* < 0.01, *** *p* < 0.001, **** *p* < 0.0001, #### *p* < 0.0001, ^^^^ *p* < 0.0001, ns: no significance.

## Data Availability

The processed data are provided in the Appendix A. Additional data supporting the results of this study are available from the corresponding author upon reasonable request.

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
