# Peer review of "PSAT1 Promotes Metastasis via p-AKT/SP1/ITGA2 Axis in Estrogen Receptor-Negative Breast Cancer Cell"

_biomolecules, 2024, doi:10.3390/biom14080990_

Round 1

Reviewer 1 Report

Comments and Suggestions for Authors

Manuscript titled “PSAT1 promotes metastasis via p-AKT/SP1/ITGA2 pathway in 2 estrogen receptor-negative breast cancer cell” by Xingda Zhang et al, is very well written with minor comments. Text is very detailed yet concise to capture the study findings. Experiments are designed appropriately to prove the hypothesis and detailed in the methods section.

Minor reviewer comments are as below:

1. Authors need to include PSAT1 protein and mRNA levels in all experimental figures where levels of its downstream effectors under PSAT1 KD or OE were reported. For example, in Figure 4D.

2. Authors need to include timepoints and concentrations in figure legends or methods. These are not mentioned anywhere in the manuscript. For example, details like how long were the cells treated with siITGA2 or MIT or LY294002 or the concentration of siITGA2 used.

3. Figure 1E shows PSAT1 high in ER negative metastatic samples while the authors report opposite in lines 181 to 183. Please correct. Also, in the same sentences, they report 85 nonmetastatic samples when there were 83 reported in other parts of the manuscript. Please report the correct number.

4. In line 20 (abstract), change “serious” to “series”.

5. Sentence in lines 297 and 298 is incomplete. Please correct.

Author Response

The responses to reviewers are listed below.

Reviewer #1: Manuscript titled “PSAT1 promotes metastasis via p-AKT/SP1/ITGA2 pathway in 2 estrogen receptor-negative breast cancer cell” by Xingda Zhang et al, is very well written with minor comments. Text is very detailed yet concise to capture the study findings. Experiments are designed appropriately to prove the hypothesis and detailed in the methods section.

Comment 1: Authors need to include PSAT1 protein and mRNA levels in all experimental figures where levels of its downstream effectors under PSAT1 KD or OE were reported. For example, in Figure 4D.

Response: We are very grateful for your comments regarding our manuscript. We added the expression of PSAT1 in Figure 4-6(Fig4D, 4G, 4E, 5D, 6A, 6C). We modified the corresponding Figure Legend and using yellow highlighting(line285,286,288,289,320-322,351-354).

Comment 2: Authors need to include timepoints and concentrations in figure legends or methods. These are not mentioned anywhere in the manuscript. For example, details like how long were the cells treated with siITGA2 or MIT or LY294002 or the concentration of siITGA2 used.

Response: We are very grateful for your comments regarding our manuscript. We have identified them in the Material and Methods and Figure Legend sections of the manuscript and using yellow highlighting. Specifically, as follows:

  1. We have labeled the plasmid concentration and siRNA concentration on line 77-79.The final concentration of the plasmid used was 5μg/ml and the siRNA transfection concentration was 50 nM.
  2. We have labeled the MIT concentration and the treatment time in line 326.
  3. We have labeled the LY294002 concentration and treatment time in line 359.

Comment 3: Figure 1E shows PSAT1 high in ER negative metastatic samples while the authors report opposite in lines 181 to 183. Please correct. Also, in the same sentences, they report 85 nonmetastatic samples when there were 83 reported in other parts of the manuscript. Please report the correct number.

Response: We are very grateful for your comments regarding our manuscript. We have corrected these two clerical errors to "more metastatic samples (33/65) had intense PSAT1 cytoplasmic staining than did nonmetastatic samples (28/83) (Fig. 1E)." In lines 182-183 of the manuscript and highlighted in yellow.

Comment 4: In line 20 (abstract), change “serious” to “series”.

Response: We are very grateful for your comments regarding our manuscript. We have corrected these mistakes.

Comment 5: Sentence in lines 297 and 298 is incomplete. Please correct.

Response: We thank the reviewers for their comments. We have revised sentences in lines 297-300 of the original manuscript. Changed to “In this study, we initially investigated the potential for PSAT1 to enhance the modulation of SP1. As depicted in Figure 5A, our results demonstrated that overexpression of PSAT1 resulted in an upregulation of SP1 expression within the nucleus.”

Reviewer 2 Report

Comments and Suggestions for Authors

The manuscript entitled “PSAT1 promotes metastasis via p-AKT/SP1/ITGA2 pathway in estrogen receptor-negative breast cancer cell’ by Zhang et al., is an interesting manuscript potentially suitable for Biomolecules. Authors have brought a self-contained story about the role of PSAT1, a phosphoserine aminotransferase 1 enzyme in regulating cell adhesion, migration and invasion of the estrogen-negative breast cancer cells via regulating the expression of ITGA2 expression in SP1 transcription factor dependent manner. I have raised following comments that should help manuscript strengthened further:

1.       The introduction part of the manuscript needs to be improved further by giving sufficient introductions of the molecules studied and interlinking them sufficiently to provide a better story and picture of the study.

2.       The term used in the study “p-Akt/SP1/ITGA2 pathway” should be explained in another way. This term seems to refer to the well-recognized signaling pathway rather than referring to the interconnected role of these molecules.

3.       Authors may explain possible mechanism in discussion part how a PAST1 kinase can regulate the nuclear translocation of the SP1 transcription factor. Could it be a direct or indirect effect?

4.       Does PSAT1 over-expression also drive enhanced adhesion, migration, invasion of ER-positive breast cancer cells?

5.       Fig.6B, for how long, authors treated cells with LY compound? Do authors also try PI 3-kinase specific inhibitors for these experiments?

6.       The model in Figure 6F looks like PSAT1 as downstream signaling molecule of the ITGA2 in membrane. Has the author tried to see whether PSAT1 interacts with ITGA2 to activate the Akt?

Author Response

The responses to reviewers are listed below. All revisions are highlighted in the manuscript.

Reviewer #2: The manuscript entitled “PSAT1 promotes metastasis via p-AKT/SP1/ITGA2 pathway in estrogen receptor-negative breast cancer cell’ by Zhang et al., is an interesting manuscript potentially suitable for Biomolecules. Authors have brought a self-contained story about the role of PSAT1, a phosphoserine aminotransferase 1 enzyme in regulating cell adhesion, migration and invasion of the estrogen-negative breast cancer cells via regulating the expression of ITGA2 expression in SP1 transcription factor dependent manner. I have raised following comments that should help manuscript strengthened further:

Comment 1: The introduction part of the manuscript needs to be improved further by giving sufficient introductions of the molecules studied and interlinking them sufficiently to provide a better story and picture of the study.

Response: We are very grateful for your comments regarding our manuscript. We have added a paragraph on the existing molecular mechanisms and molecular pathways of PSAT1 in lines 49-60 of the introduction along with references 17-20.

“A recent study found that PSAT1 also activates the Notch1/β-catenin signaling pathway to upregulate the metastatic ability of breast cancer and promote distant metastasis17. Similarly, PSAT1 interacts with IQGAP1 and subsequently promotes cell migration through STAT3 phosphorylation18. We wondered whether PSAT1 could extensively activate key proteins affecting malignant functions in tumors through phosphorylation. AKT, a well-known protein regulating GSK-3β, is frequently overactivated by phosphorylation in malignant tumors19. Moreover, AKT inhibitors are potentially effective drugs for the treatment of breast cancer20. Therefore, we propose that PSAT1 may promote metastasis in ER-negative breast cancer by regulating AKT phosphorylation. We wondered whether PSAT1 is capable of broadly activating key proteins affecting malignant tumour function through phosphorylation and ultimately promoting breast cancer metastasis.”

Comment 2: The term used in the study “p-Akt/SP1/ITGA2 pathway” should be explained in another way. This term seems to refer to the well-recognized signaling pathway rather than referring to the interconnected role of these molecules.

Response: Thank you for your comment. We also did feel that the term pathway, was more in favor of the classical molecular pathways known to the general public rather than the new molecular regulatory mechanism we proposed, so we modified “pathway” to “axis” in the title and in the keywords, and highlighted it (line2,22,27,69,338,356, 360,394,422).

Comment 3: Authors may explain possible mechanism in discussion part how a PAST1 kinase can regulate the nuclear translocation of the SP1 transcription factor. Could it be a direct or indirect effect?

Response: Thank you for your comment. We are very grateful for your comments regarding our manuscript. We have explained this issue to the Discussion (line 415-421). We believe that the indirect regulation of SP1 by PSAT1 is through p-AKT. In this study, we hypothesized that PSAT1 might be involved in the regulation of ITGA2 through the p-AKT pathway by performing KEGG pathway enrichment analysis on differentially expressed genes after overexpression of PSAT1.Overexpression of PSAT1 promoted the phosphorylation of AKT, which in turn facilitated the entry of SP1 into the nucleus and the transcriptional activation of ITGA2, which ultimately enhanced migration and invasion of ER-negative breast cancer cells. Meanwhile, inhibition of PSAT1 led to the opposite result. Thus, the PSAT1/p-AKT/SP1/ITGA2 axis may be the molecular mechanism that promotes metastasis in ER-negative breast cancer cells.

Comment 4: Does PSAT1 over-expression also drive enhanced adhesion, migration, invasion of ER-positive breast cancer cells?

Response: Thank you for your comment. We have found low levels of PSAT1 expression in estrogen receptor-positive breast cancer in our previous study, and in the current study (Fig 1E), we found that in estrogen-negative breast cancer, the levels of PSAT expression were upregulated and statistically different between lymph node metastasis and non-metastasis groups. In contrast, the same result was not found in estrogen-positive breast cancer (Fig 1F). Therefore, we hypothesized that low PSAT1 expression in estrogen receptor-positive breast cancers may lack function in breast cancer metastasis. Therefore, we did not further verify the effect on its cell adhesion, invasion and metastasis functions. In addition, it has also been reported in the literature that PSAT1 has been reported to be affected by DNA methylation in ER-positive breast cancer cells, resulting in suppression of PSAT1 expression. Thereby the downstream pathway of PSAT1 is inhibited. This is consistent with our results. Therefore, we did not explore the biological function of PSAT1 that is lowly expressed in ER-positive breast cancer cells. We have placed this section in the "Discussion line 425-433" and highlighted it.

Comment 5: Fig.6B, for how long, authors treated cells with LY compound? Do authors also try PI3-kinase specific inhibitors for these experiments?

Response: We thank the reviewers for this comment. For Figure 6B, we used different concentrations of LY294002 acted for 1h. We added this section to the figure notes (lines 379), and used highlighting. LY294002 is a PI3K-AKT pathway inhibitor and is frequently used to inhibit AKT phosphorylation [1-4], including in breast cancer [5, 6]. This study focuses on PSAT1 to promote SP1 entry into the nucleus and transcriptional activation of ITGA2 through P-AKT activation to promote breast cancer cell metastasis. Since the AKT molecule has multiple isoforms such as AKT1, AKT2 and AKT3, here, we chose LY294002 to be able to inhibit the phosphorylation of AKT without differentiating the isoforms.

  1. Zang L, Fu D, Zhang F, Li N, Ma X. Tenuigenin activates the IRS1/Akt/mTOR signaling by blocking PTPN1 to inhibit autophagy and improve locomotor recovery in spinal cord injury. J Ethnopharmacol. 2023 Dec 5;317:116841. doi: 10.1016/j.jep.2023.116841. Epub 2023 Jun 22. PMID: 37355079.
  2. Shan H, Lin Y, Yin F, Pan C, Hou J, Wu T, Xia W, Zuo R, Cao B, Jiang C, Zhou Z, Yu X. Effects of astragaloside IV on glucocorticoid-induced avascular necrosis of the femoral head via regulating Akt-related pathways. Cell Prolif. 2023 Nov;56(11):e13485. doi: 10.1111/cpr.13485. Epub 2023 Apr 26. PMID: 37186483; PMCID: PMC10623974.
  3. Zhao Y, Song K, Zhang Y, Xu H, Zhang X, Wang L, Fan C, Jiang G, Wang E. TMEM17 promotes malignant progression of breast cancer via AKT/GSK3β signaling. Cancer Manag Res. 2018 Aug 2;10:2419-2428. doi: 10.2147/CMAR.S168723. PMID: 30122991; PMCID: PMC6080873.
  4. Deng N, Zhang X, Zhang Y. BAIAP2L1 accelerates breast cancer progression and chemoresistance by activating AKT signaling through binding with ribosomal protein L3. Cancer Sci. 2023 Mar;114(3):764-780. doi: 10.1111/cas.15632. Epub 2022 Dec 1. PMID: 36308067; PMCID: PMC9986062.
  5. Ock CW, Kim GD. Harmine Hydrochloride Mediates the Induction of G2/M Cell Cycle Arrest in Breast Cancer Cells by Regulating the MAPKs and AKT/FOXO3a Signaling Pathways. Molecules. 2021 Nov 5;26(21):6714. doi: 10.3390/molecules26216714. PMID: 34771123; PMCID: PMC8588485.
  6. Xie Y, Kim HI, Yang Q, Wang J, Huang W. TRPV3 regulates Breast Cancer Cell Proliferation and Apoptosis by EGFR/AKT pathway. J Cancer. 2024 Mar 25;15(10):2891-2899. doi: 10.7150/jca.93940. PMID: 38706904; PMCID: PMC11064276.

Comment 6:The model in Figure 6F looks like PSAT1 as downstream signaling molecule of the ITGA2 in membrane. Has the author tried to see whether PSAT1 interacts with ITGA2 to activate the Akt?

Response: We thank the reviewers for this comment. In Figure6F, the position of ITGA2 molecular model is really easy to be misunderstood by the readers. In the schematic diagram, we modified the position of ITGA2 molecular model to show our results more clearly. We believe that in the cytoplasm PSAT1 regulates ITGA2 through P-AKT/SP1, and ITGA2 is a PSAT1-promoted breast cancer cells metastasis as a downstream target.

Reviewer 3 Report

Comments and Suggestions for Authors

The manuscript is interesting, however it is suggested to carefully review Wang Z, Wu Z, Cell. 2023 Sep 28;186(20):4454-4471.

 The written information regarding overexposure in breast tumors and the prognosis of metastasis is confusing. Perhaps by reviewing and comparing the data obtained with that of the already published literature, PSAT1 can be concluded more clearly and accurately.

It is suggested to review the sharpness of Fig 2 since the visibility in the tissue photographs is slightly impaired. The qPCR technology in the table does not describe the fluorophores and quencher used, it would be important to attach this information

Author Response

The responses to reviewers are listed below.

Comment 1:  The manuscript is interesting, however it is suggested to carefully review Wang Z, Wu Z, Cell. 2023 Sep 28;186(20):4454-4471.

Response: We are very grateful for your comments regarding our manuscript. We first read the reviewer-suggested article titled "An immune cell atlas reveals the dynamics of human macrophage specification during prenatal development". This study provides a comprehensive map of the heterogeneity and developmental dynamics of human macrophages and unravels their diverse functions during development. macrophages and unravels their diverse functions during development. This is an important reference for macrophage research. At present, in all honesty our research is not deep enough. Many thanks to the reviewers for giving us references for more in-depth research in the future. Thank you for your contribution.

Comment 2: The written information regarding overexposure in breast tumors and the prognosis of metastasis is confusing. Perhaps by reviewing and comparing the data obtained with that of the already published literature, PSAT1 can be concluded more clearly and accurately.

Response: We thank the reviewers for this comment. Regarding the question of correlation between tumor metastasis and prognosis, our manuscript does lack a description of tumor metastasis and prognosis. We herein plotted PSAT1 in ER-negative breast cancer in relation to patient survival using the TCGA database. The data show that in patients with ER-negative breast cancer, survival was lower in the high PSAT1 expression group than in the low PSAT1 expression group. We put the results in an additional graph (line 186-188, Fig S1A). We also found that PSAT1 expression was upregulated in the lymph node metastasis group of estrogen-negative breast cancer. Thus, high PSAT1 expression leads to breast cancer metastasis and results in poor prognosis. Meanwhile, previous studies have found that high PSAT1 expression promotes tumor progression in a variety of cancers, and we have added the corresponding content in the Introduction section (line 49-60).(To make it easier for reviewers to view the revisions, we have placed FIGS1 on the last page of this revised version of the manuscript.)

Comment 3:It is suggested to review the sharpness of Fig 2 since the visibility in the tissue photographs is slightly impaired. The qPCR technology in the table does not describe the fluorophores and quencher used, it would be important to attach this information

Response: We thank the reviewers for this comment. For the issue of Figure 2 clarity, we have increased the resolution at which Figure 2 is generated. We hope this will give the reader a better sense of visualization. For the lack of PCR using fluorophores and quencher described things. We used SYBR Green premixed reagent (ROX) produced by Roche. This kit is characterized by simplifying the PCR process i.e. PCR experiments can be carried out by adding only this reagent with cDNA and the corresponding primers without the need to add other luminescent groups and quenching agents. We have attached the link to the Roche kit (https://elabdoc-prod.roche.com/LifeScience/Document/0ff30cff-98f5-e311-98a1-00215a9b0ba8).

Round 2

Reviewer 2 Report

Comments and Suggestions for Authors

Authors have provided response to my questions.